# Woody species composition and diversity of agroforestry homegardens along altitudinal gradient in southwest Ethiopia

**Tefera Jegora**[1,2]*, **Kitessa Hundera**[1], **Zerihun Kebebew**[1], **Adugna Eneyew**[1]

**1** College of Agriculture and Veterinary Medicine, Jimma University, Jimma, Ethiopia, **2** College of Agriculture and Forestry, Mettu University, Metu, Ethiopia

* teferajegora@gmail.com

**Data Availability Statement:** All data are contained within the manuscript.

## Abstract

Homegarden agroforestry systems that integrate trees with agricultural practices are usually valued for the conservation of farm biodiversity. Despite the system having a significant conservation role, litle is known on woody species composition and diversity following the elevation belt of southwest Ethiopia. A complete enumeration of 72 homegardens (24 each from altitudinal gradient) was purposively selected for woody species inventory. A total of 55 woody species belonging to 31 families and 45 genera were recorded. Of which, 56.4% of woody species are indigenous and two are endemic to Ethiopia. Family Fabaceae was the most represented family with eight species. The highest species richness (42) was observed at high altitudes followed by 39 species at middle and 31 species at low altitudes but no significant difference between them. Species richness significantly (P < 0.001) increased with increasing wealth status. The overall richness distribution was 46, 40, and 27 across rich, medium, and poor wealth classes respectively. Shannon diversity index differed significantly between poor and rich households. Pearson correlation result shows a strong positive and significant correlation between richness and wealth status. The mean woody species density was 89.06±9.25 and 1236.22±131.42 per garden and hectare, respectively. Stem density was significantly higher (P < 0.001) in wealthier farms. We found that, in southwestern Ethiopia wealthier agroforestry farms can support more woody species diversity in homegardens.

## Background

Agroforestry is a strategic land use plan to conserve biodiversity in deforested and fragmented landscapes [1–3]. The practice integrates the most environmentally sound forestry and agricultural systems into the same land management unit with sustainable and profitable interaction between trees, crops, and livestock [4]. It supports the floral and faunal assemblages that can be as species-rich, abundant, and diverse as forests with modified and non-forest species compositions [5,6]. These help agroforestry systems to have diverse components and multi-strata structures with multiple ecological and socio-economic benefits [7–9].

**Funding:** The author(s) received no specific funding for this work.

**Competing interests:** The authors have declared that no competing interests exist.

Agroforestry is a potential biodiversity conservation site following intact natural forests as it provides an important reservoir for terrestrial biodiversity [10,11]. Tropical agroforestry systems have high levels of woody species diversity that are naturally retained and/or planted on farmland [12–14]. A diversified woody species in the homegarden agroforestry systems of southwest Ethiopia is a common practice, which is dominated by khat and coffee shrubs. *Coffea arabica* and *Catha edulis* are the two most important income-generting woody species of homegardes in southern Ethiopia [15]. It encourages tree plantation and maintenance to diversify multipurpose woody species that support smallholder farmers' livelihoods [16–18]. Agroforestry systems serve as in situ conservations containing high species richness and diversity valued and conserved by the farmers [10,19]. As a result, agroforestry plays five vital roles in biodiversity conservation: (1) provides habitat for species to tolerate a certain level of disturbance; (2) reduces the rate of conversion of natural habitat by providing more productive and sustainable alternative agricultural systems; (3) provides corridors between habitat remnants and the conservation of area-sensitive floral and faunal species; (4) preserves the germplasm of sensitive species; and (5) provides ecosystem services such as erosion control, increased infiltration capacity, reduced degradation, and habitat fragmentation [3,20–22].

Homegarden agroforestry is usually managed by members of the households for the production of subsistence crops and sometimes cash objectives [23]. The diverse and multi-strata components of the system not only provide significant food and income sources [24–27] but also serve as biodiversity conservation [28,29]. It varies from the nearby agroforestry practices due to their mosaic patches or plots dominated by a diversity of woody species [15]. Consisting of multiple strata, complex structure, high species richness, evenness, and abundance make homegardens typically similar to natural forests [6,30]. Homegarden contains high species diversity and usually 3–4 vertical canopy strata, including trees as an upper story, herbaceous vegetation as near ground, and intermediate layers in between [4,7,15]. This composite nature of the agroforestry system is under the influence of different ecological and socio-economic factors globally [14,31,32]. For example, [33] revealed that household level of resource endowment significantly affects tree number, species richness, and species diversity.

In Ethiopia, agroforestry is an old farming system containing low to high woody species diversity distribution due to variations in ecological and socio-economic factors such as elevation, size of farmland, and wealth status [15,25,34–36]. Woody species composition in the agroforestry system is influenced by household wealth status and other socio-economic attributes such as the size of farmland, level of education, family size, and off-farm economic sources [37]. In indigenous agricultural practices of the northern Ethiopian homegardens, [38] reported variations between woody species composition, density, and diversity along altitudinal gradients. However, there is no comprehensive evidence on woody species compasition and diversity from homegarden agroforestry following altitudinal and wealth status change in the present study area. Therefore, this study was conducted to reveal woody species composition and diversity of homegarden agroforestry systems along an altitudinal gradient and household wealth status in southwest Ethiopia.

## Materials and methods

### Description of the study area

The study was conducted in three districts, namely Mana, Gomma, and Dhidhessa, following the elevation belt of southwest Ethiopia. The area is located between 7°40'N-8°10'N and 36°30'E-36°50'E with an elevation range of 1459m to 2542m above sea level (Fig 1). The area received mean annual rainfall of 1853mm with minimum and maximum temperature of 11.5–24.2°C, respectively for the past 32 years [39]. It is occupied by cultivated land, wetlands,

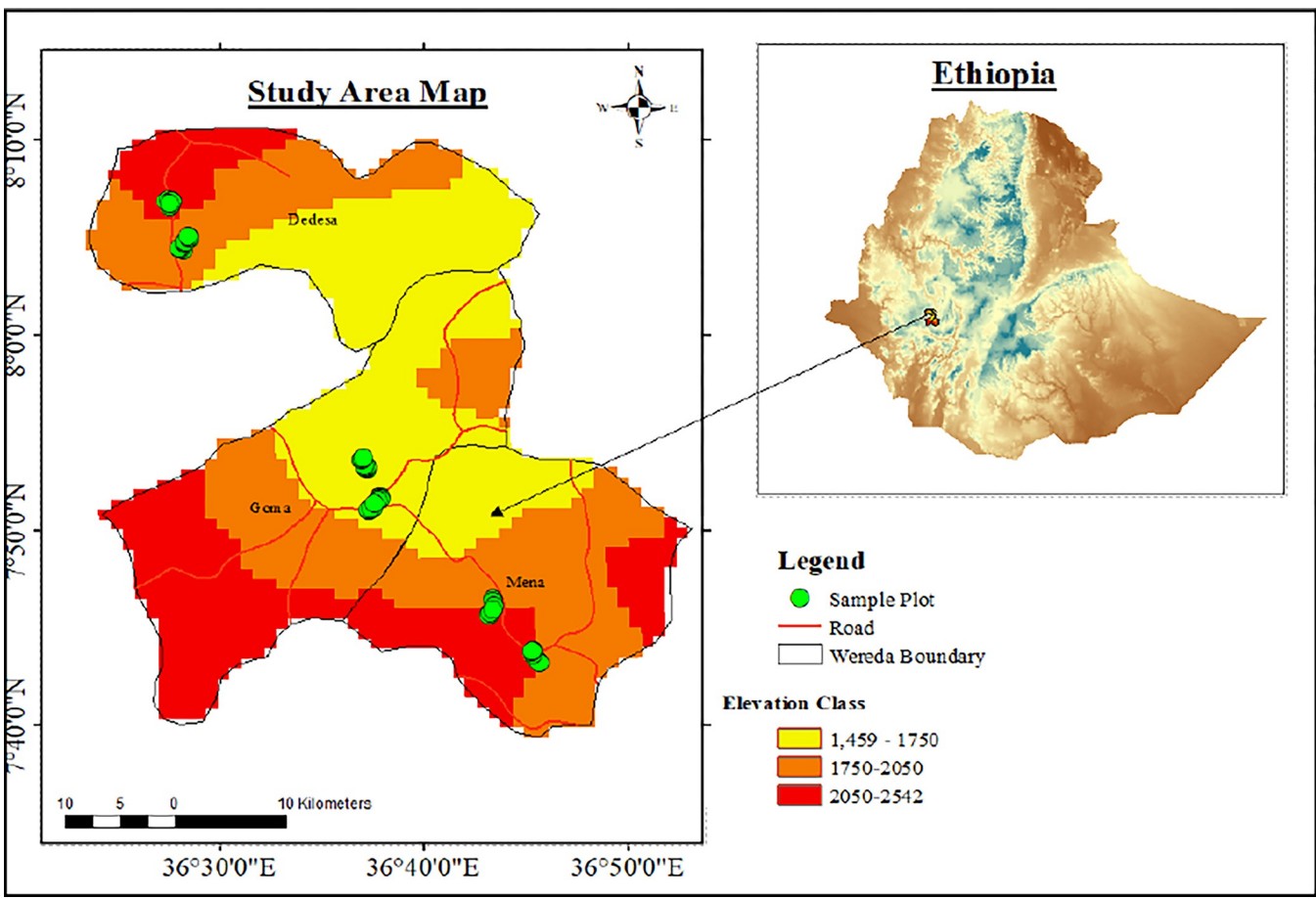

**Fig 1. Map of the study area.** The map was created using shapefiles acquired for free from the Ethiopian Mapping Agency. The Digital Elevation Model (SRTM, 30m resolution) can be accessed for free at EarthExplorer (usgs.gov).

settlements, commercial farms, shrubland, grassland, and forest land. Agro-ecologically, it is classified into lowland, midland, and highland, with midland-dominated areas. Agriculture is the main source of socio-economic development, with coffee and khat as dominant cash crops. Homegarden agroforestry systems based on coffee, enset and khat are the most accredited farming systems.

## Sampling strategy

Woody species inventory was conducted in homegarden agroforestry systems across the elevation belt of southwest Ethiopia. Elevation data was gathered from the Digital Elevation Model in Google Earth [40] and classified purposively based on the presence of homegarden agroforestry systems. Then, three altitudinal gradients were made as low (1459–1750 m), middle (1750–2050 m), and high (2050–2542 m) following the distributions of homegarden agroforestry with its respective altitude to see the effect of altitudinal change on woody species diversity. Additionally, wealth status was classified as rich, medium and poor to examine whether it correlated positively or negatively with woody species in homegarden agroforestry systems [14,41]. The classification was made with the assistance of the developmental agents and local elders. Here, the developmental agent refers to the agricultural extension worker at the smallest administrative unit (locally called kebele) in Ethiopia. Local criteria such as the number of

livestock holdings, total size of land holdings, housing standard and annual consumption of food and sell crop production were used in this classification. Then, a complete homegarden enumeration was made for the woody species inventory [42]. Finally, a total of 72 homegardens (24 from each altitudinal station) were purposively selected for woody species inventory.

## Methods of data collection

Woody species with a diameter at breast height (DBH) ≥5cm and height ≥1.5m were identified and their height and DBH were recorded. Individuals below this measurement were only counted and registered. All branches at the breast height (1.3 m) were measured for DBH and slammed for a given individual tree [36]. Tree DBH and height were measured using a caliper and a Sunto Clinometer respectively. Tree size beyond the caliper measurement was measured using mater tape and converted to diameter. Tree height difficult to measure with a Sunto Clinometer was measured by a marked range pole. The altitudinal gradient was measured using the Global Positioning System (GPS) Garmin 72. Woody species were identified on the field and voucher specimen. A voucher specimen was collected, pressed, dried and identified following [43] and flora of Ethiopia [44–46].

## Data analysis

Woody species diversity was analyzed using diversity indices [47–49]. Shannon Wiener diversity index which takes species richness and evenness into consideration was calculated as:

$$H' = -\sum_{i=1}^{s} pilnpi \tag{1}$$

Where: $H'$ = Shannon-Wiener diversity index, $s$ = number of species and $Pi$ = the proportion of individuals of the $i^{th}$ species and ln = $s$ natural logarithm.

A measure of similarity of the abundances of different woody species in the sampled area was analyzed using Evenness/Equitability index.

$$E = \frac{H'}{Hmax} = \frac{H'}{lnS} \ with \ Hmax = lnS \tag{2}$$

Where: $E$ = Evenness (Equitability), $Hmax$ = ln($S$) is the maximum level of diversity possible within a given population.

The similarity in species composition of two sites was measured using Sorensen's Similarity Coefficient ($S_s$) with values ranging between 0 and 1: 0 indicates complete dissimilarity and 1 complete similarity.

$$Ss = \frac{2a}{2a + b + c} \tag{3}$$

Where: $S_s$ = Similarity index, $a$ = number of species common in the two sites; $b$ = number of species recorded only in site one; and $c$ = number of species recorded only in site two.

Importance value index (IVI) is a useful ecological parameter to compare the environmental significance of woody species. IVI was calculated using the sum of relative density, frequency and dominance as:

$$IVI = Relative \ density + Relative \ dominance + Relative \ frequency \tag{4}$$

$$\text{Where : Relative Density (RDe)} = \frac{\text{Stem count of a species}}{\text{Stem count of all species}} *100 \qquad 4.1$$

$$\text{Relative Dominance (RDo)} = \frac{\text{Doninance of a species}}{\text{Dominance of all species}} *100 \qquad 4.2$$

$$\text{Relative Frequency (RF)} = \frac{\text{Frequency of a species}}{\text{Frequency of all species}} *100 \qquad 4.3$$

Density was calcualted as the number of individuals per ha:

$$D = N/S \qquad (5)$$

Basal area, the area occuied by woody species was calculated for all woody species DBH ≥5cm using the following formula:

$$\text{Basal area (BA)} = \pi d^2/4 \qquad (6)$$

Where: $\pi$ = 3.14; d = diameter at breast height (m)

Statistical analysis was made using R software version 4.1.3 [50]. Homogeneity and normality of the variance were assessed using the LeveneTest and Shapiro-Wilk tests in R statistics. Non-normally distributed data were log-transformed and outliers were removed. The variables were compared using one-way ANOVA. The relationship between altitudinal gradients and wealth status and the diversity indices was examined using Pearson's correlation coefficient test. The least significant difference (LSD) test was used to compare the level of significance ($P < 0.05$) between each variable.

## Results

### Floristic composition

In total, 55 woody species belonging to 31 families and 45 genera were recorded in homegarden agroforestry systems. Family Fabaceae was the most dominant having eight species followed by Moraceae, Myrtaceae and Rutaceae each with four species. About 34.5% of the remaining families had only one woody species. About 56.4% of species are indigenous including two endemics (*Erythrina brucei* and *Millettia ferruginea*), and the remaining 43.6% are exotic (S1 Appendix 1 in S1 File). The values of species richness were 31, 39 and 42 for low, middle and high altitudes, respectively. Woody species richness increased with increasing wealth status. Accordingly, 27, 40 and 46 species were recorded for poor, medium and rich households respectively (Table 1). The differences in species richness were not significant ($P < 0.05$) between altitudinal gradients, whereas it significantly ($P < 0.001$) increased with increasing wealth status (Table 2). The percentage frequency occurrence of woody species lay between 1.4–84.7%. *Coffea arabica* (84.7%), *Persea americana* (79.5%), and *Catha edulis* (54.2%) are the most frequently observed species. However, about 15 species were observed only once in surveyed homegardens (S2 Appendix 2 in S1 File).

### Woody species diversity

The Shannon diversity and evennes indices were not significantly different ($P < 0.05$) between altitudinal gradients. The highest mean Shannon (1.07±0.09) and evenness (0.57±0.04) indices were observed at middle altitude. The lowest Shannon (0.97±0.08) was recorded at high

**Table 1. Overall woody species richness, genera and family distribution of woody species across altitudinal gradient and wealth status in homegarden agroforestry systems.**

| Category | Number | Richness | Genus | Family |
|---|---|---|---|---|
| **Altitudinal gradient** | | | | |
| **Low** | 24 | 31 | 26 | 21 |
| **Middle** | 24 | 39 | 36 | 25 |
| **High** | 24 | 42 | 37 | 25 |
| **Total** | 72 | 55 | 45 | 31 |
| **Wealth class** | | | | |
| **Poor** | 24 | 27 | 24 | 21 |
| **Medium** | 24 | 40 | 33 | 24 |
| **Rich** | 24 | 46 | 39 | 27 |
| **Total** | 72 | 55 | 45 | 31 |

altitudes, whereas the lowest evenness (0.53±0.03) was recorded at low altitudes. Species richness and mean Shannon diversity index increased with household wealth status, whereas evenness showed the reverse. The Shannon diversity index was significant (P < 0.05) between poor and rich households, while evenness didn't show a significant difference between wealth (Table 2). There was no correlation between species richness and altitude, but a positively significant correlation was between species richness and wealth status. Furthermore, the Shannon diversity index correlated significantly and positively with wealth status (Table 3).

## Woody species similarity

About 51% of the woody species were the same along the altitudinal gradient and across wealth classes (Fig 2). The highest similarity index was observed between high and middle altitudes (55%) and the lowest (45%) was registered between high and low altitudes. For a wealth class, the highest (60%) similarity was seen between high and medium classes while the lowest (40%) was seen between medium and poor classes (Table 4). On the other hand, 12, seven and three woody species were unique to high, middle and low altitudes respectively, whereas 10, six and two unique specie were recorded for rich, medium and poor households respectively (Fig 2).

**Table 2. Species richness, Shannon diversity index and Evenness (Mean ± SE) of woody species across altitudinal gradients and wealth status in southwest Ethiopia.**

| Category | Richness | Shannon | Evenness |
|---|---|---|---|
| **Altitudinal gradients** | | | |
| **Low** | 7.54[a]±0.59 | 1.00[a]±0.06 | 0.53[a]±0.03 |
| **Middle** | 7.42[a]±0.77 | 1.07[a]±0.09 | 0.57[a]±0.04 |
| **High** | 7.75[a]±0.76 | 0.97[a]±0.08 | 0.55[a]±0.05 |
| **Mean** | 7.57±0.40 | 1.02±0.05 | 0.55±0.02 |
| **Wealth status** | | | |
| **Poor** | 4.75[c]±0.42 | 0.84[b]±0.06 | 0.61[a]±0.04 |
| **Medium** | 7.58[b]±0.46 | 1.07[a]±0.09 | 0.53[a]±0.04 |
| **Rich** | 10.38[a]±0.66 | 1.14[a]±0.07 | 0.50[a]±0.03 |
| **Mean** | 7.57±0.40 | 1.02±0.05 | 0.55±0.02 |

Key- Different letters superscripted along column mean value indicate significant different (P < 0.05), SE- standard error.

**Table 3. Pearson correlation between altitudinal gradient, wealth status and woody species diversity in homegarden agroforestry systems.**

| Category | Altitudinal gradients | Wealth status | Species richness | Shannon diversity index |
|---|---|---|---|---|
| **Wealth status** | 0.000 | - | - | - |
| **Species richness** | 0.025 | 0.674** | - | - |
| **Shannon diversity index** | -0.028 | 0.319** | 0.470** | - |
| **Evenness** | 0.055 | -0.225 | -0.270* | 0.610** |

Note:

*- significant different (P < 0.05)

**- significant different (P < 0.01)

## Woody species density

The mean stem density per homegarden and hectare was 89.06±9.25 and 1236.22±131.42 respectively. The mean stem density was significantly different (P < 0.001) between wealth classes. The highest 134.88±18.87 stem density was recorded in the rich followed by medium 87.08±14.35 and poor 45.21±7.38 wealth classes. However, there was no significant difference between stem density and altitudinal gradient at the garden and hectare level (Table 5).

## Basal area distribution

The mean basal area was highest at middle altitude followed by high and low altitude respectively, although there was no significant difference between each altitude. Homegarden level basal area was 1.53±0.29 for the rich, 0.90±0.27 for the medium, and 0.33±0.08 for the poor in which a significant difference (P < 0.01) was seen between the poor and rich classes (Table 5). However, there was no significant difference in basal area per hectare (m$^2$ ha$^{-1}$) along elevation

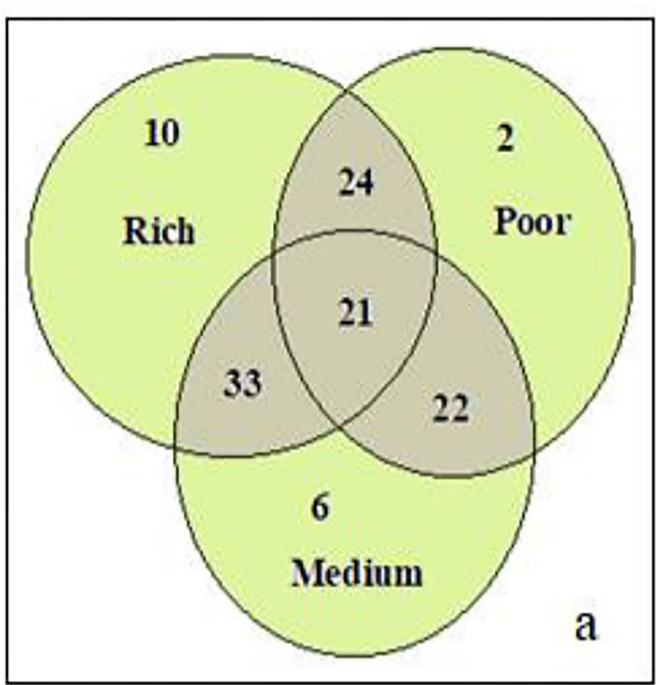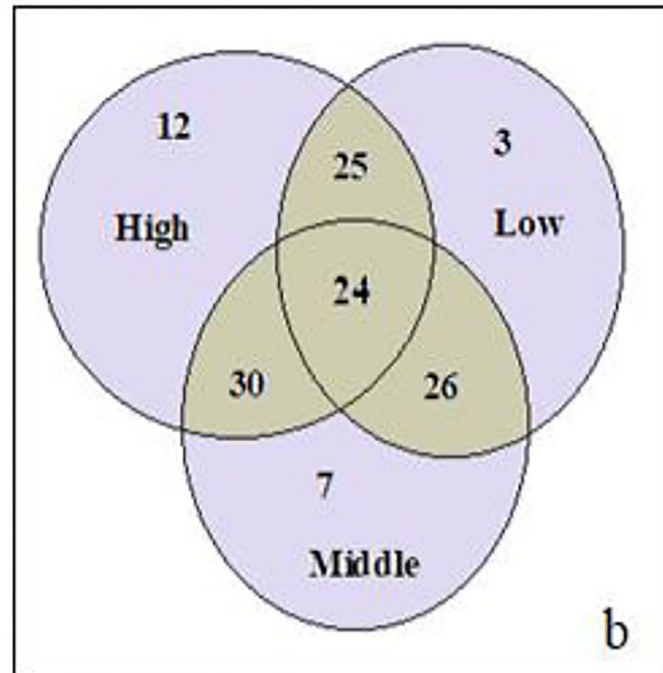

**Fig 2.** Venn diagram representation of the similarity and dissimilarity of woody species across (a) wealth status and (b) altitudinal gradient in homegarden agroforestry systems.

**Table 4. Woody species similarity index in homegarden agroforestry systems across altitudinal gradient and wealth status.**

| Altitudinal gradient | High | Middle | Low |
|---|---|---|---|
| Middle | 0.55 | - | - |
| Low | 0.45 | 0.47 | - |
| Wealth status | Rich | Medium | Poor |
| Medium | 0.60 | - | - |
| Poor | 0.44 | 0.40 | - |

gradients and wealth status. Finally, the overall basal area distributions of the top ten woody species are presented in Fig 3. *Persea americana* had the highest basal area (886.87 m² ha⁻¹) followed by *Cordia africana* (146.48 m² ha⁻¹), *Coffea arabica* (130.87 m² ha⁻¹), *Mangifera indica* (95.68 m² ha⁻¹), *Catha edulis* (64.98 m² ha⁻¹), *Grevillea robusta* (46.30 m² ha⁻¹), *Citrus sinensis* (18.01 m² ha⁻¹), *Psidium guajava* (16.79 m² ha⁻¹), *Carica papaya* (15.69 m² ha⁻¹) and *Albizia gummifera* (6.51 m² ha⁻¹) (Fig 3).

## Importance value index

The overall importance value index in the entire homegarden agroforestry system value lay between 0.20 to 69.06% for less to more important woody species respectively. However, a large number (29) of woody species had an importance value index of less than 1% (S1 Appendix 1 in S1 File). Fig 4 indicates the importance value index distribution of the top ten woody species. Accordingly, the importance value of these species was presented as *Catha edulis* (69.06%), *Coffea arabica* (51.01%), *Persea americana* (40.84%), *Cordia africana* (18.07%), *Mangifera indica* (15.81%), *Grevillea robusta* (11.98%), *Psidium guajava* (9.31%), *Citrus sinensis* (9.17%), *Ricinus communis* (7.97%) and *Carica papaya* (6.87%) in their descending order (Fig 4).

## Discussion

Woody species composition and diversity significantly vary between wealth statuses but not between altitudinal gradients. Species richness and the Shannon diversity index have shown a significant increasing trend with increasing wealth status. The result was agreed with [14,51], who reported increased richness and Shannon diversity index with increasing wealth status in south-central Ethiopia. Evenness had shown a reverse trend with richness in wealth classes. The result indicates that homegardens with few numbers of different species (lower species

**Table 5. Mean (±SE) woody density and basal area distribtution along altitudinal gradients and wealth status in homegarden agroforestry systems.**

| Category | Density/garden | Density/ha | BA/garden | BA/ha |
|---|---|---|---|---|
| **Altitudinal gradients** | | | | |
| Low | 82.04ᵃ±14.44 | 1151.88 ᵃ±140.63 | 0.64ᵃ±0.13 | 8.97ᵃ±1.49 |
| Middle | 79.88ᵃ±12.10 | 1298.50ᵃ±275.63 | 1.28ᵃ±0.31 | 15.00ᵃ±3.49 |
| High | 105.25ᵃ±20.48 | 1258.29ᵃ ±252.41 | 0.84ᵃ±0.28 | 11.29ᵃ±3.79 |
| **Wealth status** | | | | |
| Poor | 45.21ᶜ±7.38 | 1654.29ᵃ±268.98 | 0.33ᵇ±0.08 | 10.59 ᵃ±2.71 |
| Medium | 87.08ᵇ±14.35 | 1046.13ᵃᵇ±169.67 | 0.90ᵃᵇ±0.27 | 11.37 ᵃ±1.70 |
| Rich | 134.88ᵃ±18.87 | 1008.25ᵇ±217.68 | 1.53ᵃ±0.29 | 12.09 ᵃ±3.65 |

Key: Different letters superscripted along column mean value indicate significant different (P < 0.05), SE- standard error.

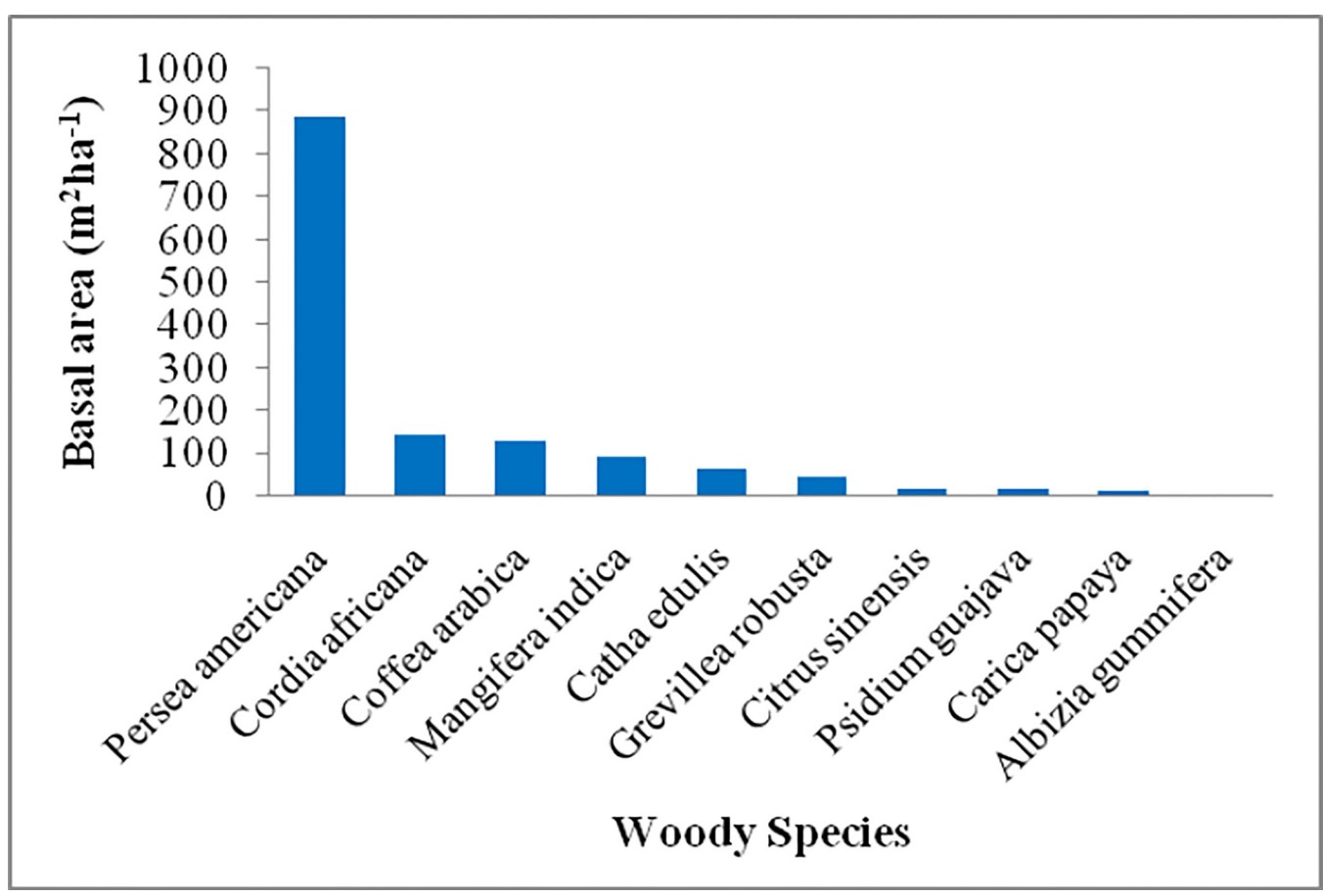

**Fig 3. Basal area distribution (m$^2$ ha$^{-1}$) for the top ten woody species in homegarden agroforestry systems at southwest Ethiopia.**

richness) have a large abundance for each species (higher evenness). However, there was no continuous increasing or decreasing values of diversity indices seen along altitudinal gradients. Likewise, species richness and diversity in agroforestry didn't show a parallel relationship with the elevation gradient [52]. *Catha edulis*, *Coffea arabica* and *Persea americana* are the most abundant and frequently recorded woody species. The study conducted by [38] reported that the distributions of farm woody species are determined by the economic or ecological values they provide. Therefore, the main reason for the higher abundance and distribution of these woody species may be due to the farmer's focus on the production of highly economic and ecologically valued woody species in the present study area.

The number of woody species recorded in the present study area is comparable to the number of woody species (55) reported in the traditional agroforestry practices of Dellomenna district, Ethiopia [53]. The number is lower than 120 trees and shrubs species reported from southern Ethiopia [15] and 60 woody species around Jimma Town, southwestern Ethiopia [39] within similar agroecology. It's also lower than 83 plant species in eastern Amazonia [54], 138 in south-central Ethiopia [14], 171 in central Nepal [55], and 206 in southern and southwestern Ethiopia [29]. However, it's higher than 53, 36, 49 and 32 woody species reported from Shewarobit district [56], Shashemene district [57], southern Ethiopia [58], and southern Tigray [28] in Ethiopia, respectively. It is also higher than 32 woody species in western Cameroon [59] and 30 woody species in the mid-highlands of southern Ethiopia [52].

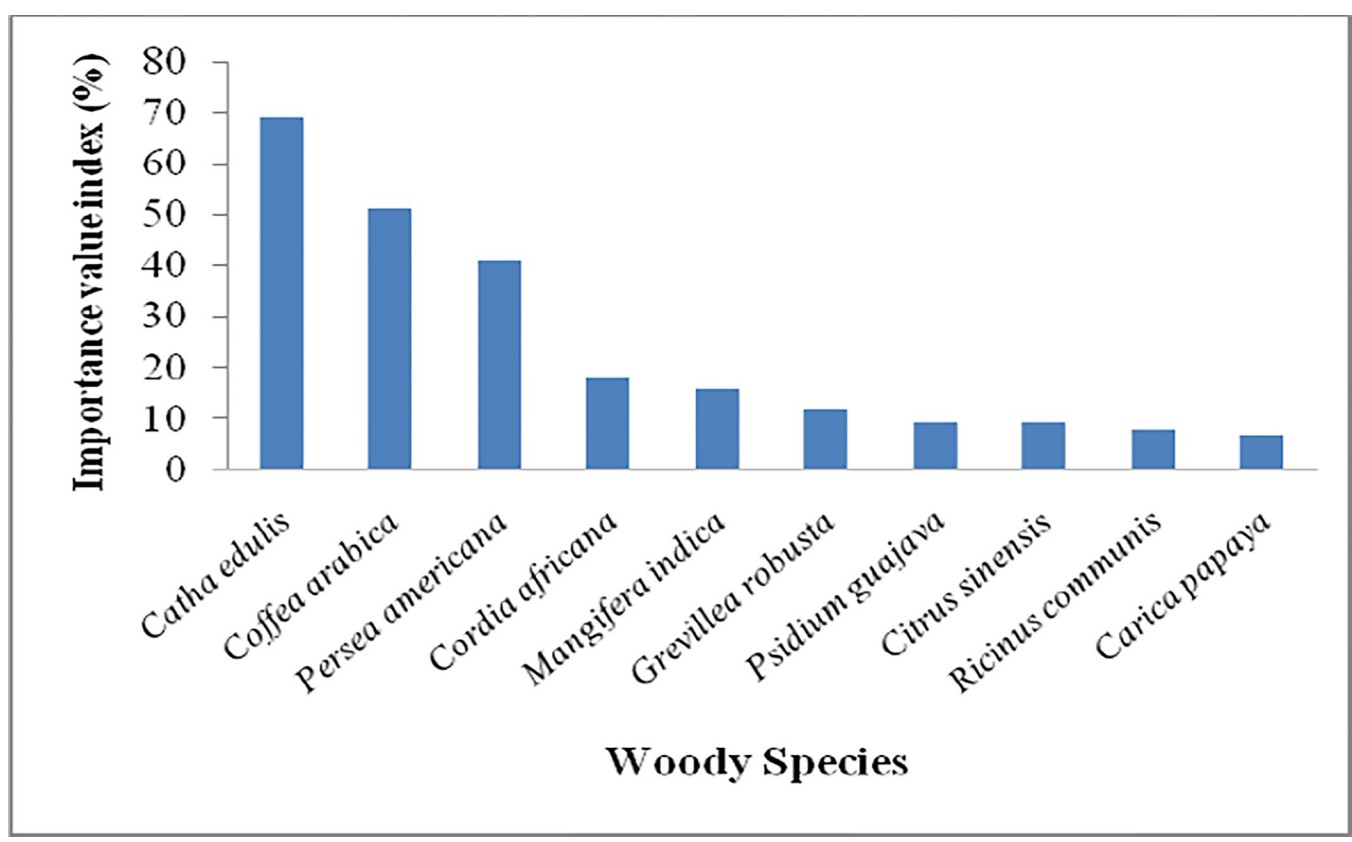

**Fig 4. Importance value index of the top ten woody species across the entire homegarden agroforestry systems.**

The present study registered 56.4% indigenous and 43.6% exotic species whereas 85% indigenous and 15% exotic species were reported in Dellomenna district, southeastern Ethiopia [53]. This inverse result may be observed due to two main reasons. 1) Ecological difference between the two sites as Dellomenna is located in the forest-savannah transition zone of southern Ethiopia and the present study area is located in Eastern Afromontane Forests dominated by woody plantation following the expansion of farmland in southwest Ethiopia. 2) Variation in farmers' preference ranking towards the selection of an exotic and native woody species. The present finding results agreed with the study conducted by [29] who reported 69.52% indigenous and 31.48% exotic in homegarden agroforestry systems from similar agroecology. Similar to this study, *Erythrina brucei* and *Millettia ferruginea* were also reported as the two endemic woody species in Ethiopia in traditional agroforestry practices [60].

The variation in the woody species similarity index recorded was not only because of elevation change but also wealth status most probably related to the farmer's homegarden size, extension service, improved seedlings, and labor forces. Likewise, the species similarity index between the sites changed with elevation and woody production [61]. The highest similarity index reported between high and middle altitudes agreed with [52], who reported the highest species similarity between adjacent elevations in coffee-based agroforestry in Ethiopia. Also, [59] reported the highest woody species similarity due to closed ecological conditions between coffee agroforestry systems and forests in Cameroon.

The mean stem density significantly increased with wealth status but there was no significant difference between altitudinal gradients. This indicates that household wealth status

determines woody species density in homegarden agroforestry systems. The present study reported the top ten woody species having the highest basal area. Of which, *Persea americana*, an exotic fruit tree species has the highest (61.52%) basal area. However, [51] reported that *Eucalyptus camaldulensis* had the highest (30.03%) basal area among the top ten woody species basal area registered in traditional agroforestry practices of Wondo district, south-central Ethiopia. The distribution indicates that *Cordia africana*, a native endangered tree occupies the second largest basal area which has been recorded on the fourth step among woody species having the highest importance value index. The highest importance value index was recorded for *Catha edulis*. Similarly, *Catha edulis* was recorded as an economically important woody crop in homegardens [62]. On the contrary, *Eucalyptus camaldulensis* [15,51], *Faidherbia albida* [34], and *Cordia africana* [40] were reported as the most important woody species in homegarden agroforestry systems in Ethiopia.

## Conclusion

Homegarden agroforestry systems in southwestern Ethiopia are rich in woody species. Households within a rich wealth class had the highest number of woody species compared to medium and poor wealth classes. Woody species richness was not statistically significant between altitudinal gradients. Shannon diversity index significantly increases with increasing wealth status, with no significant difference between altitudinal gradients. The finding indicates that household economic status determines the diversification of woody species in ecologically related human-managed land use systems. Therefore, to improve woody species diversification in homegarden agroforestry, concerned bodies should support the residents through the provision of appropriate supplies. Furthermore, more investigation will be required to determine the specific variables influencing the woody species composition and diversity in homegarden agroforestry and its sustainable socio-economic and ecological uses.

## Supporting information

**S1 File.**
(DOC)

## Author Contributions

**Conceptualization:** Tefera Jegora, Kitessa Hundera, Zerihun Kebebew, Adugna Eneyew.

**Formal analysis:** Tefera Jegora.

**Investigation:** Tefera Jegora.

**Supervision:** Kitessa Hundera.

**Validation:** Zerihun Kebebew, Adugna Eneyew.

**Writing – original draft:** Tefera Jegora.

**Writing – review & editing:** Kitessa Hundera, Zerihun Kebebew, Adugna Eneyew.

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
