## [Decision Letter · Decision Letter 0]

15 Apr 2024

PONE-D-24-12537Woody species composition and diversity of homegarden agroforestry along altitudinal gradient in southwest EthiopiaPLOS ONE

Dear Dr. Kepura,

Thank you for submitting your manuscript to PLOS ONE. After careful consideration, we feel that it has merit but does not fully meet PLOS ONE’s publication criteria as it currently stands. Therefore, we invite you to submit a revised version of the manuscript that addresses the points raised during the review process.

We look forward to receiving your revised manuscript.

Kind regards,

Narel Y. Paniagua-Zambrana, M.D.

Academic Editor

PLOS ONE

Journal Requirements:

2. In the online submission form, you indicated that [Data will be available upon the request].

Additional Editor Comments (if provided):

The reviewers have made comments and suggestions that should be taken into account by the authors for the submission of a new version of their paper.

Reviewers' comments:

Reviewer's Responses to Questions

**Comments to the Author**

1. Is the manuscript technically sound, and do the data support the conclusions?

Reviewer #1: Yes

Reviewer #2: Yes

2. Has the statistical analysis been performed appropriately and rigorously? 

Reviewer #1: Yes

Reviewer #2: Yes

3. Have the authors made all data underlying the findings in their manuscript fully available?

Reviewer #1: Yes

Reviewer #2: Yes

4. Is the manuscript presented in an intelligible fashion and written in standard English?

Reviewer #1: No

Reviewer #2: No

5. Review Comments to the Author

Reviewer #1: This manuscript reports interesting results, among them that the woody species richness and stem density are higher in wealthier agroforestry farms. The methods of data collection and analysis seem to be appropriate, so the results must be credible.

However, I regret to report that this manuscript is not ready for reviewing, and the problem is its language. I find your writing style very unnatural, far from being strictly academic, sometimes not easy to comprehend. As an example, I added my suggestions to the Title and the Abstract, however, I am neither able nor qualified to edit your entire text. Please find a native English speaking colleague or a professional editor who understands this area (Ethnobotany, agroscience, ecology), your text requires an extensive editing. After that it can be properly reviewed.

A minor problem is formatting. The Abstract is in italics, some tables are in italics… it would be easier to read if only species names are italicized.

Numbers in the text not always need brackets.

There are no line numbers, they would be handy for locating the commented bits of text.

Reviewer #2: The authors present a manuscript dealing with homegardens in a still underreported region in Ethiopia.

Overall the analysis is clear. However, the ms does have a variety of problems that need to be addressed:

LANGUAGE:

The English needs to be adjusted to international publishing standards.

GENERAL

Introduction is a bit fragmentary and does not well introduce into the subject overall, nor the local conditions in particular. Thus the introduction needs to be expanded.

RESULTS

"Erythrina brucei and Millettia ferruginea"- yes these are endemic to Ethiopia but are of Least concern because rather widepread. Erythrina species are common in agroforestry around the globe

"1.4 - 84.7%. Coffea arabica (84.7%), Persea americana (79.5%), and Catha edulis (54.2%)" - this means the systems were largely plantations of these species

"Persea americana had the highest basal area (886.87 m2 ha-1) followed by Cordia africana (146.48 m2 ha-1), Coffea arabica (130.87 m2 ha-1), Mangifera indica (95.68 m2 ha-1), Catha edulis (64.98 m2 ha-1), Grevillea robusta (46.30 m2 ha-1), Citrus sinensis (18.01 m2 ha-1), Psidium guajava (16.79 m2 ha-1), Carica papaya (15.69 m2 ha-1) and Albizia gummifera (6.51 m2 ha-1) (Fig. 3)." - here it would be interesting to point out that TWO NATIVE SPECIES (Cordia and Albizzia) actually have a high basal area. Same for the high importance value of Cordia africana. Also. COFFEA ARABICA is of course also a native species that still forms natural stands in Ethiopia.

"Catha edulis, Coffea arabica and Persea americana are the most abundant and frequently recorded woody species. The study conducted by Gebrewahid & Abrehe (2019) reported that the distributions of farm woody species are determined by the economic or ecological values they provide. Therefore, the main reason for the higher abundance and distribution of these woody species may be due to the farmer’s focus on the production of highly economic and ecologically valued woody species in the present study area" - YES INDEED! Which, sadly, means that this study does not present much new information.

"The present study registered 56.4% indigenous and 43.6% exotic whereas 85% indigenous and 15% exotic reported in Dellomenna district, southeastern Ethiopia (Molla & Kewessa, 2015)." - YES - but it is not discussed WHY. Dellomenna is in the forest-savannah transition zone in southern Ethiopis. So the differences between the two locations should be discussed. Also, exactly for that reason, lots of the exotic species planted in the study region will not grow well in Dellomenna.

The

"Appendix 1: List of woody species scientific, local and family name, and their origin in homegarden agroforestry systems of southwest Ethiopia

No Scientific name Local name Family name Origin

1 Acacia abyssinica Hochst. ex Benth. Sondii Fabaceae I

2 Albizia gummifera (J. F. Gmel.) C.A. Sm."

IS A BIT MESSY. Authorities MUST NOT be in italics. It should be

Acacia abyssinica Hochst. ex Benth. AND SO ON. REVISE ALL NAMES

DISCUSSION

Discussion ignores a large part of the pertinent comparative literature and shopuld be expanded

6. PLOS authors have the option to publish the peer review history of their article (what does this mean?). If published, this will include your full peer review and any attached files.

Reviewer #1: No

Reviewer #2: No

---

## [Author Response · Author response to Decision Letter 0]

7 May 2024

All issues raised by the reviewer and editor were presented as described in detail in the attached documents.

---

## [Decision Letter · Decision Letter 1]

4 Jun 2024

PONE-D-24-12537R1Woody species composition and diversity of agroforestry homegardens along altitudinal gradient in southwest EthiopiaPLOS ONE

Dear Dr. Kepura,

Thank you for submitting your manuscript to PLOS ONE. After careful consideration, we feel that it has merit but does not fully meet PLOS ONE’s publication criteria as it currently stands. Therefore, we invite you to submit a revised version of the manuscript that addresses the points raised during the review process.

We look forward to receiving your revised manuscript.

Kind regards,

Narel Y. Paniagua-Zambrana, M.D.

Academic Editor

PLOS ONE

Journal Requirements:

Additional Editor Comments:

I appreciate that you can consider reviewers' comments and suggestions to improve your manuscript.

Reviewers' comments:

Reviewer's Responses to Questions

**Comments to the Author**

1. If the authors have adequately addressed your comments raised in a previous round of review and you feel that this manuscript is now acceptable for publication, you may indicate that here to bypass the “Comments to the Author” section, enter your conflict of interest statement in the “Confidential to Editor” section, and submit your "Accept" recommendation.

Reviewer #1: All comments have been addressed

Reviewer #2: All comments have been addressed

2. Is the manuscript technically sound, and do the data support the conclusions?

Reviewer #1: Yes

Reviewer #2: Yes

3. Has the statistical analysis been performed appropriately and rigorously? 

Reviewer #1: Yes

Reviewer #2: Yes

4. Have the authors made all data underlying the findings in their manuscript fully available?

Reviewer #1: Yes

Reviewer #2: Yes

5. Is the manuscript presented in an intelligible fashion and written in standard English?

Reviewer #1: Yes

Reviewer #2: Yes

6. Review Comments to the Author

Reviewer #1: I found all my previous comments incorporated and altogether the manuscript is much improved. However, some weird wordings still can be found along with some typos, so the text still will need certain edition.

The most salient result of this work is that species richness and abundance (stem density) are found in the home gardens of wealthier farmers. It is not a novelty, but the study seems to be conducted in an important region where this type of research has not been done. This point needs more clarification in the Introduction where the authors state their aims and objectives.

Also, there is no comparison of the area occupied by the home gardens: do home gardens of the wealthier farmers occupy more land than those of poorer farmers? This important data is lacking. That there are significant differences per ha in the stem and species densities can indicate that it is not just the larger land area that contains more plants/species, but also that the wealthier farmers select trees and manage their gardens more carefully.

In other relations — data collection and analysis, presentation of the results and their discussion — the manuscript does not present any serious problem. The structure of the manuscript is clear, all relevant references are cited, the conclusions are based on the results of statistical tests.

These and my specific comments are given in the attached file.

Reviewer #2: The authors have made a honest effort to improve their manuscript and address all the issues raised by the reviewers. Overall this has been successful, and the new version is much improved., There are still some minor language issues that should be addressed before publication, but this might be done in the proofing stage.

7. PLOS authors have the option to publish the peer review history of their article (what does this mean?). If published, this will include your full peer review and any attached files.

Reviewer #1: No

Reviewer #2: No

---

## [Author Response · Author response to Decision Letter 1]

3 Jul 2024

We have seen all the comments given by the reviewers. It is a clear and constructive idea to improve the quality of the manuscript. Therefore, we address all the comments raised by the reviewers as ordered by the journal editor-in-chief. Finally, we inform you that we have no problems with the given comments and have prepared the revised version and attached it in a separate file (as Manuscript).

---

## [Decision Letter · Decision Letter 2]

28 Oct 2024

Woody species composition and diversity of agroforestry homegardens along altitudinal gradient in southwest Ethiopia

PONE-D-24-12537R2

Dear Dr. Kepura,

We’re pleased to inform you that your manuscript has been judged scientifically suitable for publication and will be formally accepted for publication once it meets all outstanding technical requirements.

Kind regards,

Narel Y. Paniagua-Zambrana, M.D.

Academic Editor

PLOS ONE

Additional Editor Comments (optional):

We appreciate the work done by the authors to improve the quality and content of the manuscript. We consider that it is acceptable for publication.
---

## [Editor Report · Acceptance letter]

4 Nov 2024

PONE-D-24-12537R2 

PLOS ONE

Dear Dr. Jegora, 

I'm pleased to inform you that your manuscript has been deemed suitable for publication in PLOS ONE. Congratulations! Your manuscript is now being handed over to our production team.

Kind regards, 

on behalf of

Dr. Narel Y. Paniagua-Zambrana 

Academic Editor

PLOS ONE